# Successful Live Twin Birth through IVF/ICSI from a Couple with an Infertile Father with Pericentric Inversion of Chromosome 9 (p12q13): A Case with a High Aneuploidy Rate

**DOI:** 10.3390/medicina58111646

**Published:** 2022-11-14

**Authors:** Ning-Shiuan Ting, Ying-Hsi Chen, Shih-Fen Chen, Pao-Chu Chen

**Affiliations:** 1Department of Obstetrics and Gynecology, Hualien Tzu Chi Hospital, Buddhist Tzu Chi Medical Foundation, Hualien 970, Taiwan; 2Reproductive Health and IVF Center, Hualien Tzu Chi Hospital, Buddhist Tzu Chi Medical Foundation, Hualien 970, Taiwan

**Keywords:** aneuploidy, infertility, in vitro fertilization, pericentric inversion, chromosome 9

## Abstract

Evidence suggests that the pericentric inversion of chromosome 9 (inv(9)) does not affect the aneuploidy rate (38.5%) after IVF. Herein, we report a successful live female twin birth through IVF/ICSI with a high aneuploidy rate from a couple within which the infertile father has inv(9)(p12q13). A couple (a 34-year-old male and a 35-year-old female) was referred to our clinic due to infertility. The wife has a child with her previous husband. Results from the infertility workup of both parents were normal. Karyotyping revealed that the inv(9)(p12q13) of the father was the only cytogenetic abnormality. Preimplantation genetic testing for aneuploidies (PGT-A) after IVF/ICSI revealed a high aneuploidy rate (77%; 10/13). Two euploid blastocysts were transferred, resulting in a successful live female twin birth. The presented case highlights the possibility that inv(9)(p12q13) in males may impact the fertility and euploidy rate. PGT-A facilitates the selection of qualified blastocysts for the optimization of live-birth outcomes.

## 1. Introduction

The pericentric inversion of chromosome 9 (inv(9)) is one of the most common variants of the human karyotype and has a frequency ranging from 1–4% in the general population [1,2]. The inv(9) has been accepted as a benign population variant with no clinical significance [3], but sustained debate remains in the literature; some studies reported the association of inv(9) with reproductive disorders, such as infertility and miscarriage [4,5,6], whereas other studies showed contradictory results [2,7,8]. Although abnormal gametogenesis has been hypothesized to be involved [9], the mechanisms that predispose inv(9) carriers to reproductive disorders remain unclear.

Recently, emphasis has been directed to investigating the association of inv(9) with reproductive outcomes after in vitro fertilization and/or intracytoplasmic sperm injection (IVF/ICSI). Two studies reported that male inv(9) carriers [1] or inv(9) carriers, including both sexes [10], were not inferior to noncarriers in terms of major reproductive outcomes. Notably, a study [11] reported that the aneuploidy rate of inv(9) patients was not significantly higher than that of the controls (38.5% and 40.7%, respectively), suggesting that this population variant is not associated with increased reproductive risks in assisted reproductive technology. In this report, we present a case of a successful live birth of female twins through IVF/ICSI with a high aneuploidy rate from a couple in which the infertile male partner has inv(9) (p12q13).

## 2. Case Report

A couple (a 34-year-old male and a 35-year-old female) was referred to our fertility clinic on June 13, 2018, because of two years of infertility. The couple was phenotypically normal and nonconsanguineous. The pedigree evaluation showed neither congenital anomalies nor infertility problems in the family. The female partner had a child with her previous male partner and was presumed to not have issues of infertility. The results from her infertility workup were normal. The present male partner also had normal findings from the semen analysis, physical examination, and urogenital examination (testicular volume, serum follicle-stimulating hormone, testosterone, and prolactin) [12]. The data of the morphology, motility, total count, and level of DNA fragmentation of sperm analysis were all within the normal ranges. For patients with unexplained infertility, further karyotype analysis regarding potential chromosome defects has been recommended [13]. Cytogenetic analysis revealed a karyotype of 46,XY,inv(9)(p12q13) (Figure 1) for the male partner and a normal karyotype for the female partner. Owing to multiple intrauterine insemination failures and the advanced age (age > 34 years) of the wife, they underwent IVF/ICSI with next-generation sequencing-based preimplantation genetic testing for aneuploidies (PGT-A).

The female partner underwent transvaginal oocyte retrieval five times within two years, and 33 mature oocytes were retrieved. Without oocyte freezing, after IVF/ICSI, 25 zygotes were obtained (fertilization rate: 73%), of which 24 zygotes were two pronuclei (2PN), and one zygote was three pronuclei (3PN). Among the 24 zygotes, 14 progressed to blastocysts. Subsequently, a trophectoderm cell biopsy was performed on days 5 or 6. The embryos were frozen after the biopsy through vitrification procedures [14]. PGT-A analysis revealed that only three blastocysts were classified as euploid, and 10 blastocysts were classified as aneuploid (Table 1). The aneuploidy rate was as high as 77%. No result was obtained from the remaining blastocyst because of whole genome amplification failure. Among the three euploid blastocysts, one had developmental arrest after thawing and cultivation. Two euploid blastocysts with good quality were then transferred to the uterus, resulting in successful twin pregnancies. An amniocentesis test was performed in the second trimester, and the result showed that both fetuses had normal karyotypes. The female partner finally gave birth to normal dizygotic twin baby girls at 36 weeks of pregnancy through cesarean section delivery.

## 3. Discussion

This report describes a case of a successful live birth of female twins through assisted reproductive technology from a couple in which the infertile male partner has inv(9) (p12q13). After IVF/ICSI, the high aneuploidy rate (77%) resulted in a situation in which only two euploid blastocysts were qualified for transfer. The live-birth outcome of this case was highly satisfactory.

The association between inv(9) and infertility remains controversial [2,4,5,6,7]. The inv (9)(p12p13) variant is frequently detected in Chinese infertile couples [1,10]. Given that the wife has a child from her previous marriage and the results from the infertility workup of both parents were normal, the inv(9) (p12q13) of the father was the only cytogenetic abnormality for the unexplained infertility in our case. These findings seem to support the notion that inv(9) may have an impact on fertility [4,5,6].

The impact of inv(9) on reproductive outcomes after IVF/ICSI has attracted considerable interest. Among these outcomes, the euploidy rate is a crucial factor closely related to the success rate of assisted reproductive technology [15]. Young et al. [9] reported that the aneuploidy rate (41.7%) in 163 embryos from 36 inv(9) patients was similar to that (47.2%) from a group of maternal age-matched controls. Additionally, they found no inherited unbalanced rearrangements in the embryos and suggested that chromosome inversions are not susceptible to interchromosomal effects during meiosis in blastocysts. Merrion and Maisenbacher [11] reported that the aneuploidy rate of inv(9) patients was not significantly higher than that of the controls (38.5% and 40.7%, respectively) and concluded that this population variant is not associated with increased reproductive risks. In our case, we observed a high aneuploidy rate. The reason for this high aneuploidy rate remains unclear. The involvement of abnormal gametogenesis in the vulnerability of inv(9) carriers to reproductive failure has been proposed previously [9].

Apart from the aneuploidy rate, other reproductive outcomes after IVF/ICSI are clinically important. Li et al. [1] reported that rates of fertilization, 2PN cleavage, quality embryo, implantation, and clinical pregnancy in the inv(9) group of male carries (n = 25) did not significantly differ from those in the control group (n = 1088). Only lower rates of fertilization and 2PN cleavage were found in the inv(9) group of female carriers (n = 37). The inv(9) group of both sexes had higher rates of early miscarriage. Liang et al. [10] compared various IVF/ICSI outcomes, including rates of utilization, clinical pregnancy, implantation, live-birth rate, and miscarriage, of a group composed of 107 couples with inv(9), with the control group composed of 107 couples with normal karyotypes. They [10] reported that the inv(9) group did not show any disadvantages when compared with the control group. The findings from these studies [1,10] suggest that inv(9) has an unremarkable impact on reproductive outcomes after IVF/ICSI.

In our case, we performed PGT-A to exclude embryos with chromosomal abnormalities [16] and to select the best embryo in order to optimize the live-birth rate per transfer. In addition to the conventional morphologic score, the genetic status of embryos identified by PGT-A has been associated with the success rate of assisted reproductive technology [15,17]. The application of PGT-A in assisted reproductive technology has not been well defined [15,17]. Some researchers recommended the use of PGT-A for improving reproductive outcomes [9,10,11,15], but others reported results that do not support the benefits of PGT-A, such as in subfertile women [17] or in chromosomal inversion carriers, except for inversions on chromosome 9 [18]. In view of the high aneuploidy rate and the highly satisfactory live-birth outcome in our case, we think that the use of PGT-A is well justified.

## 4. Conclusions

Whether inv(9)(p12q13) in males may impact the fertility and euploidy rate after IVF/ICSI remains to be elucidated. In our case, a high aneuploidy rate decreases the likelihood of the success of assisted reproductive technology. PGT-A can facilitate the selection of qualified blastocysts for the optimization of live-birth outcomes. Appropriate genetic counseling is required for infertile patients. Future studies may be directed to investigate the common characteristics of the aneuploidy blastocysts that may explain the origin of the aneuploidy.

## Figures and Tables

**Figure 1 medicina-58-01646-f001:**
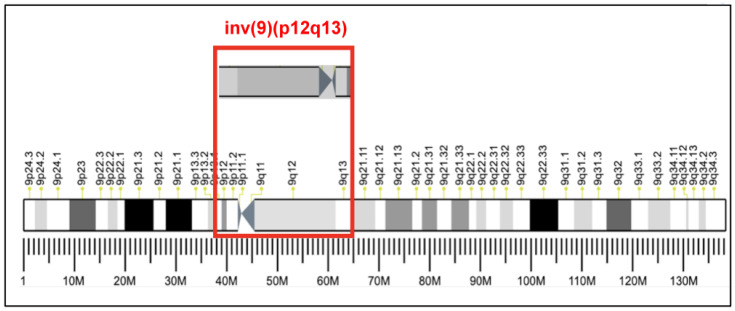
Cytogenetic map of the pericentric inversion of chromosome 9 (p12q13) in the infertile male partner.

**Table 1 medicina-58-01646-t001:** Findings of preimplantation genetic testing for aneuploidies (PGT-A) and conditions of euploid blastocysts. Next generation sequencing (NGS)-based PGT-A was performed. PGT-A, preimplantation genetic testing for aneuploidy; NGS, Next generation sequencing.

Blastocyst Number	PGT-A Result	Karyotype	NGS Report or Conditions of Blastocysts	Related Diseases
1	Aneuploid	45,XY,−s8,−15	Partial deletion of chromosome 8 Monosomy of chromosome 15	1. Maybe related to Multiple Myeloma 2. Prader-Willi Syndrome (PWS) is 15q11—13 of deletion
2	Aneuploid	47,XY,+13	Trisomy of chromosome 13	Patau’s syndrome
3	Aneuploid	45,XY,−12	Monosomy of chromosome 12	Fatal
4	Aneuploid		Multiple chromosomal abnormalities	Fatal
5	Euploid	46,XX	Qualified for transfer	-
6	Euploid	46,XX	Qualified for transfer	-
7	Aneuploid	45,XX,−7	Monosomy of chromosome 7	Myelodysplastic Syndromes
8	No result		Whole genome amplification failure	-
9	Aneuploid	46,XY,−s4,+s4,−s6,+s6	Partial deletion and duplication of chromosome 4 and 6	Fatal
10	Aneuploid	51,XX,+9,+13,+14,+18,+20	Trisomy of chromosome 9, 13, 14, 18 and 20	Fatal
11	Aneuploid		Multiple chromosomal abnormalities	Fatal
12	Euploid	46,XX	Developmental arrest after thawing and cultivation	-
13	Aneuploid	47,XY,+13	Trisomy of chromosome 13	Patau’s syndrome
14	Aneuploid		Multiple chromosomal abnormalities	Fatal

## Data Availability

All data shown in this study are included in the published article.

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
