# Peer review of "Successful Live Twin Birth through IVF/ICSI from a Couple with an Infertile Father with Pericentric Inversion of Chromosome 9 (p12q13): A Case with a High Aneuploidy Rate"

_medicina, 2022, doi:10.3390/medicina58111646_

Round 1
Reviewer 1 Report
The authors presented the case of the twin live birth following IVF tretament and PGT-A in a couple with the male parter carrying the inv(9) (p12q13).
1. As the male parter showed normal sperm analysis and physical examination (and hormonal profile?) the indication for the karyotype analysis is actually missing. Please clarify the diagnostic workflow of the male partner.
2. "Cytogenetic analysis revealed a karyotype of 46,XY,inv(9) (p12q13) of the father and a normal karyotype of the mother". This sentence is confousing. Actually the authors are talking about the male and female partner, respectively.
3. "The mother underwent transvaginal oocyte retrieval five times within 2 years". Did the authors perform multiple collection via oocyte freezing or blastocyst biopsy in each cycle? Please clarify
3. Due to normale reproductive outcomes in vitro (fertilization rate and blastilation rate) the putative negative impact on fertility should be exclude in the present couple. On the other hand, the abnormally high aneuploidy rate should be considered with caution, particularly in comparison to other studies in literature. Please clarify this attitude in discussion
Line 63. Biopsy was performed
Author Response
Responses to reviewer 1
We would like to thank the reviewers for their extensive assessment of our manuscript, and for important and helpful comments and suggestions. We have responded to all the reviewer’s comments in a point-by-point fashion and have revised the manuscript accordingly. The revised portions are indicated by “Track Changes”. We hope that the changes made will be considered satisfactory.
Comment 1. As the male partner showed normal sperm analysis and physical examination (and hormonal profile?) the indication for the karyotype analysis is actually missing. Please clarify the diagnostic workflow of the male partner.
Response 1: We thank the reviewer for the suggestion. We indeed went through the step-by-step diagnostic workflow sequentially including history and clinical evaluations [12]. The normal results from these tests suggested that the male had unexplained infertility. For this category of patients, further karyotype analysis regarding potential chromosome defects has been recommended [13]. In response to the suggestion, we have added a statement to describe this recommendation of further karyotype analysis in patients with unexplained infertility (line 57-62). A reference has been added to support this statement.
Comment 2: "Cytogenetic analysis revealed a karyotype of 46,XY,inv(9) (p12q13) of the father and a normal karyotype of the mother". This sentence is confousing. Actually the authors are talking about the male and female partner, respectively.
Response 2: We thank the reviewer for the suggestion. The sentence has been modified to avoid confusion and it reads as:”Cytogenetic analysis revealed a karyotype of 46,XY,inv(9) (p12q13) (Fig. 1) of the male partner and a normal karyotype of the female partner.” (line 61-62)
Comment 3: "The mother underwent transvaginal oocyte retrieval five times within 2 years". Did the authors perform multiple collection via oocyte freezing or blastocyst biopsy in each cycle? Please clarify
Responses 3: We did not perform oocyte freezing after transvaginal oocyte retrieval. In response to the suggestion, the sentence has been modified to provide this in formation. The statement (lines 76-80) reads as: “Without oocyte freezing, after IVF/ICSI, 25 zygotes were obtained (fertilization rate: 73%), of which 24 zygotes were two pronuclei (2PN), and one zygote was three pronuclei (3PN). Among the 24 zygotes, 14 progressed to blastocysts. Subsequently, trophectoderm cells biopsy were performed on days 5 or 6.”
Comment 4. Due to normale reproductive outcomes in vitro (fertilization rate and blastilation rate) the putative negative impact on fertility should be exclude in the present couple. On the other hand, the abnormally high aneuploidy rate should be considered with caution, particularly in comparison to other studies in literature. Please clarify this attitude in discussion
Response 4: We fully agree with the viewpoints from the reviewer. In response to the suggestion, we have changed our attitude toward the putative negative impact on fertility.
(1) In the sentence (lines 123) of “The reason for this high aneuploidy rate remains unclear, but it may be due to the negative impact of inv(9) on the euploidy rate”, we have deleted “but it may be due to the negative impact of inv(9) on the euploidy rate.”
(2) In the sentence (lines 164-165) of “Inv(9) (p12q13) in males may have negative impact on fertility and euploidy rate after IVF/ICSI”, the wording has been revised to “Whether Inv(9) (p12q13) in males may impact the fertility and euploidy rate after IVF/ICSI remains to be elucidated.”
Additionally, we have now removed the term “unusually” throughout the text.

Reviewer 2 Report
The paper is scientific sundness, without any novelty to the current literature.
The data are presented inaccurately:
1. The IVF/ICSI methodology used with immature oocytes is not described (line 59 60).
Explain better. 2. The authors reported three euploid blastocysts (line 66) obtained while in the table 1
they reported only two. Please report the correct data.
3. Please specify better the "advanced maternal age" reported in line 54
The low number of blastocyst analysed (n=13) does not allowed to conclude the "unusually high aneuploidy rate" (line 98).
Author Response
Responses to reviewer 2
We would like to thank the reviewers for their extensive assessment of our manuscript, and for important and helpful comments and suggestions. We have responded to all the reviewer’s comments in a point-by-point fashion and have revised the manuscript accordingly. The revised portions are indicated by “Track Changes”. We hope that the changes made will be considered satisfactory.
Comment 1: The paper is scientific sundness, without any novelty to the current literature.
Response 1: We thank the reviewer for his/her time and suggestion to enhance the quality of our work.
Comment 2: The IVF/ICSI methodology used with immature oocytes is not described (line 59 60).
Response 2: We apologized for the error we made. The term “33 immature oocytes” in that sentence should be “33 mature oocytes.” We have now corrected this error (line 76).
Comment 3: The authors reported three euploid blastocysts (line 66) obtained while in the table 1 they reported only two. Please report the correct data.
Response: We apologized for the confusion. We did report three euploid blastocysts obtained (line 81). Table 1 shows that PGT-A results indicated 3 euploid embryos (#5, 6, and 12), but the #12 embryo had developmental arrest after thawing and cultivation. Thus, only two euploid embryos (#5 and 6) were deemed as qualified for transfer.
Comment 4: Please specify better the "advanced maternal age" reported in line 54
Response 4: In response to the suggestion, we have specified advanced maternal age (lines 62-63). The statement read as:” Owing to multiple intrauterine insemination failures and advanced age (age > 34 years) of the wife.”
Comment 5: The low number of blastocyst analysed (n=13) does not allowed to conclude the "unusually high aneuploidy rate" (line 98).
Response 5: The suggestion from the reviewer has been well taken. We have now removed the term “unusually” throughout the text.

Reviewer 3 Report
The present manuscript shows up the problem of the pericentric inversion of chromosome 9 (inv9) related its impact on fertility and euploidy rate. Furthermore, successfully they report the case of the birth of female twins from a couple with a fertile father with inv(9). In my opinion, this case report looks interesting, as it shows a different result comparing what it has been published before in terms of inv(9) parents reproductive ability, which could be an interesting point also for other readers and scientist in the field. There are however, few suggestions that could enrich the manuscript and I think are important.
-Do the researchers found something unusual in the sperm parameters that could be related to the inv(9) variant? There is very little description of parents fertility characteristics, and enriching this point could be interesting also to have more details to compare with other studies in the bibliography, not only the fact that they have the inversion. For example, is it related to the sperm DNA integrity or something like that?
- Have the scientists of the research carried out any further analysis with the aneuploidy blastocyst to try to find any common characteristic in all of them? Something that can explain the origin of the aneuploidy? (target genes, proteins or something…)
-Have the researchers study if the aneuploid embryos where male or female? Could it be related to the fact that both implanted embryos where females?
I think all this ideas or comments should be included, if not as a possible experiment to complete the results section, at least in the discussion.
Finally, I also think that maybe the references could be extended and more actualized.
Author Response
Responses to reviewer 3
We would like to thank the reviewers for their extensive assessment of our manuscript, and for important and helpful comments and suggestions. We have responded to all the reviewer’s comments in a point-by-point fashion and have revised the manuscript accordingly. The revised portions are indicated by “Track Changes”. We hope that the changes made will be considered satisfactory.
Comment 1: Do the researchers found something unusual in the sperm parameters that could be related to the inv(9) variant?
Response 1: I thank the reviewer for reminding us this important issue. In response to the reviewer’ suggestion, we have added a statement in the revised manuscript to report the sperm parameters measured from the male partner (lines 57-59). The statement reads as: “The data of the morphology, motility, total count, and level of DNA fragmentation of sperm analysis were all within the normal ranges. “
Comment 2: There is very little description of parents fertility characteristics, and enriching this point could be interesting also to have more details to compare with other studies in the bibliography, not only the fact that they have the inversion. For example, is it related to the sperm DNA integrity or something like that?
Response 2: We thank the reviewer for the suggestion. We have now presented more detail information regarding the fertility characteristics of both parents (lines 53-59). The statements read as:” The wife has a child with her previous husband and was presumed to not having the issues of infertility. The husband also had normal findings from the semen analysis, physical examination, and urogenital examination (testicular volume, serum follicle-stimulating hormone, testosterone, and prolactin). The data of the morphology, motility, total count, and level of DNA fragmentation of sperm analysis were all within the normal ranges.” These findings regarding the fertility characteristics of both parents lead to our speculation that the cytogenetic abnormality (inversion) could be the only possibility for the unexplained infertility in our case.
Comment 3: Have the scientists of the research carried out any further analysis with the aneuploidy blastocyst to try to find any common characteristic in all of them? Something that can explain the origin of the aneuploidy? (target genes, proteins or something…)
Response 3: The suggestion from the reviewer is intriguing. However, the preimplantation genetic testing for aneuploidies revealed that these blastocysts had fatal or serious congenital defects and thus would not be processed further clinically. In response to the suggestion, we have now presented these abnormal conditions in table 1 (page 3). The suggestion from the reviewer regarding common characteristics in these aneuploidies would be an interesting topic that warrants future study.
Comment 4: Have the researchers study if the aneuploid embryos where male or female? Could it be related to the fact that both implanted embryos where females?
Response 4: We thank the reviewer for reminding this important issue. The sex of each euploid or aneuploid embryos has now been presented in table 1 (page 3). It appears that the sex was not the issue.
Comment 5: I think all this ideas or comments should be included, if not as a possible experiment to complete the results section, at least in the discussion.
Response 5: The suggestions from the reviewer have been well taken particularly regarding future study in common characteristics of the aneuploidy blastocysts. We have added a statement in the discussion section (lines 166-168) and it reads as:”Future study may be directed to investigate the common characteristics of the aneuploidy blastocysts that may explain the origin of the aneuploidy.”
Comment 6: Finally, I also think that maybe the references could be extended and more actualized.
Response 7: We thank the reviewer for the suggestion. We have added 2 valuable references to enhance the quality of the background. The references are [14] and [16].

Round 2
Reviewer 1 Report
The authors provided a revised version of the manuscript improved according to the raised issues. It can be further considered for publication.
Author Response
We thank the reviewer for the positive feedback and valuable suggestions to enhance our work.
Reviewer 2 Report
The authors made the required revision of the paper and they described better the IVF/ICSI methods.
Author Response
In response to the reviewer’s suggestion, we have added statements in the revised manuscript to describe better the IVF/ICSI methods we used (lines 67-78). These statements read as:” For IVF/ICSI, control ovarian hyperstimulation was performed using gonadotropins (Gonal-F, Elonva, Pergoveris; icryobank, Taipei, Taiwan) and antagonists of gonadotropin-releasing hormone (GnRH) (Cetrotide; icryobank, Taipei, Taiwan). Once the biggest follicle reached 12–13 mm in diameter, which was observed regularly by ultrasound scan, human chorionic gonadotropin or GnRH agonist (Lupron; icryobank, Taipei, Taiwan) was used when the diameters of three leading follicles reached 18 mm. After 36 hours, transvaginal ultrasound-guided oocyte retrieval was carried out using a 16-gauge ovum aspiration double lumen needle (K-OPSD-1635-A-L; COOK MEDICAL, Bloomington, IN, USA) at an aspiration pressure of 5 mmHg. Matured oocytes were inseminated by IVF or ICSI accordingly. Subsequently, fertilization was examined 18 hours after IVF or ICSI for the occurrence and number of pronuclei. Embryos with two pronuclei were cultured in a global medium (LifeGlobal, Guilford, CT, USA).”